# Detoxification of Aflatoxins in Fermented Cereal Gruel (*Ogi*) by Probiotic Lactic Acid Bacteria and Yeasts with Differences in Amino Acid Profiles

**DOI:** 10.3390/toxins15030210

**Published:** 2023-03-10

**Authors:** Kolawole Banwo, Taiwo Adesina, Olubunmi Aribisala, Titilayo D. O. Falade

**Affiliations:** 1Department of Microbiology, University of Ibadan, Ibadan 200132, Oyo State, Nigeria; 2International Institute of Tropical Agriculture, Ibadan 200001, Oyo State, Nigeria

**Keywords:** aflatoxin, detoxify, probiotic, lactic acid bacteria, yeasts, *Aspergillus*, toxigenic, atoxigenic

## Abstract

Toxigenic members of *Aspergillus flavus* contaminate cereal grains, resulting in contamination by aflatoxin, a food safety hazard that causes hepatocellular carcinoma. This study identified probiotic strains as aflatoxin detoxifiers and investigated the changes to the grain amino acid concentrations during fermentation with probiotics in the presence of either *A. flavus* La 3228 (an aflatoxigenic strain) or *A. flavus* La 3279 (an atoxigenic strain). Generally, higher concentrations (*p* < 0.05) of amino acids were detected in the presence of toxigenic *A. flavus* La 3228 compared to the atoxigenic *A. flavus* La 3279. Compared to the control, 13/17 amino acids had elevated (*p* < 0.05) concentrations in the presence of the toxigenic *A. flavus* compared to the control, whereas in systems with the atoxigenic *A. flavus* 13/17 amino acids had similar (*p* > 0.05) concentrations to the control. There were interspecies and intraspecies differences in specific amino acid elevations or reductions among selected LAB and yeasts, respectively. Aflatoxins B1 and B2 were detoxified by *Limosilactobacillus fermentum* W310 (86% and 75%, respectively), *Lactiplantibacillus plantarum* M26 (62% and 63%, respectively), *Candida tropicalis* MY115 (60% and 77%, respectively), and *Candida tropicalis* YY25, (60% and 31%, respectively). Probiotics were useful detoxifiers; however, the extent of decontamination was species- and strain-dependent. Higher deviations in amino acid concentrations in the presence of toxigenic La 3228 compared to atoxigenic La 3279 suggests that the detoxifiers did not act by decreasing the metabolic activity of the toxigenic strain.

## 1. Introduction

Cereals such as maize, millet, and sorghum are nutritionally important foods in sub-Saharan Africa due to their rich nutrients in carbohydrate, proteins, lipids, and vitamins [1,2]. *Ogi* is a traditional fermented cereal gruel made from maize, sorghum, or millet, which is regularly taken in Nigeria and West Africa. It is as an important element of the diet for children and adults, where it is familiar as a weaning food for infants and is the preferred food for the convalescent and elderly. *Ogi* is given unique names depending on its consistency and form when it is consumed such as *akamu*, *agidi* (*eko*), and *koko* [3]. Fermentation of cereals by probiotic microorganisms has nutritional benefits including increased digestibility, improvement in the amino acid content, reduced energy requirement for cooking due to the metabolism of complex molecules by the fermenting organisms, and improved organoleptic properties [4,5]. Additionally, probiotic organisms are known to be beneficial for the digestive system as they positively affect the microbiota in the digestive tract. Consequently, cereals provide several nutritional benefits to consumers [6]. 

Cereals are however susceptible to mycotoxin contamination including aflatoxins. People and animals living in the developing countries are especially at high risk of di-etary exposure to mycotoxins due to environmental and systemic pre-disposing condi-tions. Aflatoxins are major toxicants that are associated with impediments to the normal growth and development of children, immunosuppression, gastrointestinal disorders, and they are a cause of hepatocellular carcinoma; in animals, they also increase mortality and reduce productivity [7,8]. As a result of these negative impacts, aflatoxin management is critical. Pre- and post-harvest management strategies have been reported. These include the use of less susceptible hybrids, good agricultural practices to reduce crop stress, the use of biological control to modulate the population of naturally occurring toxigenic species, the use of hermetic storage mechanisms to prevent aflatoxin biosynthesis, and the sorting of grains to remove contaminated grains from the lot and predictive modeling [9,10,11,12]. However, it is not always possible to completely exclude the aflatoxins; consequently, detoxification of aflatoxins is important [5,13,14].

Yeasts, such as *Candida*, *Saccharomyces*, and lactic acid bacteria, such as *Lactobacillus*, *Lactococcus*, *Leuconostoc* and *Bifidobacterium,* have been reported as probiotics in fermented food products [15,16,17] that are also useful for aflatoxin detoxification [5,18]. They are also reported to increase nutritive properties such as amino acids, vitamins, minerals, and proximate contents [17,19]. However, the understanding of how nutritive properties vary in systems containing toxigenic Aspergilli or biocontrol (atoxigenic) Aspergilli has not been investigated. In this study, fermented cereal gruel, referred to as *ogi* or *akamu* and made traditionally in Nigeria and other West African countries [20,21], was investigated. Gruels from maize (*ogi*) and sorghum (*ogi baba*) were studied for aflatoxin decontamination and amino acid profiles. The ability of naturally occurring probiotic yeasts and lactic acid bacteria to detoxify aflatoxins B1, B2, G1, and G2, and their nutritional impact in the presence of either toxigenic or atoxigenic strains of *A. flavus* in controlled fermentation was investigated.

## 2. Results

### 2.1. LAB and Yeast Populations

The LAB populations increased during the fermentation time in all four forms of *ogi* from 24 h to 96 h. The total LAB population after 24 h of fermentation was 1.07 × 10^11^ cfu mL^−1^ in millet *ogi*, 1.04 × 10^11^ cfu mL^−1^ in sorghum *ogi*, 4.8 × 10^10^ in white maize *ogi*, and 5.2 × 10^10^ in yellow maize *ogi*. At the end of the 96 h fermentation period, the LAB in millet, sorghum, white maize, and yellow maize *ogi* had reached 1.56 × 10^11^, 1.46 × 10^11^, 1.12 × 10^11^, and 1.21 × 10^11^ cfu mL^−1^, respectively. Although the LAB populations were higher in millet and sorghum *ogi*, the rate of the LAB population increase was higher in maize *ogi*. The exponential increase in the LAB populations after 96 h of fermentation resulted in an overall population increase of 152%, 133%, 46%, and 40% from the populations recorded at 24 h of fermentation in yellow maize, white maize, millet, and sorghum *ogi*, respectively. In millet, the LAB population increased every 24 h by 20%, 11%, and 10%, after 48 h, 72 h, and 96 h of fermentation time, respectively. In sorghum, the LAB populations increased every 24 h by 13%, 14%, and 8%, after 48 h, 72 h, and 96 h of fermentation time, respectively. In white maize *ogi*, the LAB populations increased every 24 h by 25%, 32%, and 42%, after 48 h, 72 h, and 96 h of fermentation time, respectively. In yellow maize *ogi*, the LAB populations increased every 24 h by 44%, 25%, and 41%, after 48 h, 72 h, and 96 h fermentation time, respectively.

The total yeast population in millet increased by 64%, 102%, and 20%, between 24 h and 48 h, 48 h and 72 h, and 72 h and 96 h of fermentation, respectively. In sorghum, the total yeast population increased by 68% from 24 h to 48 h, by 50% from 48 h to 72 h, and by 20% from 72 h to 96 h. At the end of the 96 h fermentation process, the yeast populations in millet had reached 1.12 × 10^11^ cfu mL^−1^ and in sorghum they reached 1.15 × 10^11^ cfu mL^−1^. In millet, after 96 h of fermentation, the populations tripled the populations measured after 24 h, and had doubled in sorghum.

### 2.2. LAB Isolates and Their Identification

Twenty-five isolates each identified as LAB were isolate from sorghum, millet, white maize, and yellow maize. Putative identification was based on their biochemical test being negative for KOH, catalase, and oxidase tests, and all were Gram positive. All LAB isolated were able to utilize glucose but they differed in their abilities to utilize other compounds. The percentage utilization was as follows: sucrose, 57%; galactose, 79%; fructose, 79%; D-sorbitol, 43%; mannitol, 20%; and lactose, 76%. The LAB isolates were classified into the genera *Lactobacillus* (*Lb.*), *Lactiplantibacillus* (*Lp.*), *Leuconostoc* (*Leuc.*), *Limosilactibacillus* (*Lm.*), and *Lacticaseibacillus* (*Lc.*)*. Lc. casei*, *Lb. delbruecki*, *Lm. fermentum*, *Lp. plantarum*, and *Leuc. mesenteroides* were isolated from all four types of *ogi*. *Lb. acidophilus* and *Lb. paraplantarum* were isolated from maize (both yellow and white) only; *Lb. leichmanii* and *Lb. pentosus* were isolated from millet and sorghum only; while *Lb. brevis* was isolated from millet, sorghum, and yellow maize only. The most frequently isolated LAB (at 17%) were *Lb. delbrueckii* and *Leuc. mesenteroides*, while the least isolated was the *Lb. leichmanii* (at 3%). Sugar utilization patterns generally differed across the species except for *Lb. acidophilus* and *Lb. brevis,* which had similar patterns. Within the species, sugar and sugar alcohol utilization patterns were also similar, except for in *Lc. casei*, *Lm. fermentum*, *Lb. paraplantarum,* and *Lp. plantarum,* where there were rare differences (see Table 1).

### 2.3. Yeast Isolates and Their Identification

Twenty-five isolates each identified as yeasts were retrieved from market-sourced millet, sorghum, white maize, and yellow maize *ogi*. Based on the results from the observation of the cell shape, appearance, elevation, margin, occurrence, position of buds, and sugar utilization, which were used for putative identification. All yeast cells were able to utilize glucose; none of the isolated yeasts utilized d-sorbitol, mannitol, or lactose. The yeast isolates differed in their abilities to utilize the other sugars. The utilization of sucrose, galactose, and fructose was 35%, 35%, and 65%, respectively. *Candida krusei* utilized only glucose, but none of the other compounds. *Saccharomyces cerevisiae* and *C. tropicalis* utilized glucose, sucrose, galactose, and fructose.

The yeast species identified were *Candida krusei*, *C. tropicalis*, and *Geotrichum candidum*. *Candida krusei* was only able to utilize glucose, but not any of the other sugars or sugar alcohols. *C. tropicalis* was able to utilize glucose, sucrose, galactose, and fructose, but not lactose, D-sorbitol, or mannitol. *Geotrichum candidum* was able to utilize glucose and fructose, but not sucrose, lactose, galactose, D-sorbitol, or mannitol (Table 2).

### 2.4. Molecular Characterization

The sequence results were analyzed and compared to existing sequences using the BLAST tool. The percentage similarities were 100% with the strains deposited in the GenBank. Consequently, based on their sequence data in comparison with the data from the biochemistry and physiology, the isolates were identified into the genus level. Species classification was performed based solely on the sequence data from the molecular characterization. The following selected strains and their accession numbers are as follows: the strain W310 from white maize is *Limosilactobacillus fermentum* W310 with the accession no. MW811201; the strain M26 from millet is *Lactiplantibacillus plantarum* M26 with the accession no. MW811200; the strain MY115 from millet is *Candida tropicalis* MY115 with the accession no. MW811203; and the strain YY25 from the yellow maize is *Candida tropicalis* YY25 with the accession no. MW811204.

### 2.5. Probiotic Potentials and Safety Assessment of the LAB and Yeasts Strains

The tolerance of the LAB and yeast strains to different concentrations of bile salts varied, as shown in Figure 1. *Lactiplantibacillus plantarum* M26 and *Limosilactobacillus fermentum* W310 had the highest percentage tolerance of 97.3% and 89.8%, while *Candida tropicalis* YY25 and *C. tropicalis* MY115 displayed the highest tolerance of 96.1% and 81.4% to a 0.3% bile salt concentration, respectively.

The values are presented as the mean ± S.D values. The bars represent the mean and standard deviations of the independent experiments.

The ability of the LAB and yeast strains to effectively reduce the pH of their respective growth medium from a starting pH of 6.54 over a period of 48 h was assessed via an acidification test, and the results are presented in Table 3. *Lactiplantibacillus plantarum* M26 was observed to have reduced the pH of the growth medium to 3.65 while *Candida tropicalis* MY115 reduced it to 4.26 after 48 h, respectively. The safety assessment on the LAB and yeast employing DNase and gelatinase tests were negative while the hemolytic reaction exhibited gamma hemolysis, thus establishing that they are safe, as shown in Table 3.

### 2.6. Aflatoxin Concentration and Amino Acid Changes Due to LAB and Yeast Fermentation

#### 2.6.1. Aflatoxin Concentration Changes

Selected strains of LAB and yeasts based on their superior characteristics of bile tolerance, safety, and antioxidant scavenging were taken further for assessment on their aflatoxin decontamination potential. Expectedly, LAB and yeast isolates that were co-inoculated with the atoxigenic *A. flavus* did not result in any changes in the aflatoxin concentration since the isolate is a non-aflatoxigenic *A. flavus* strain. However, there were changes in the amino acid profiles, which will be discussed hereafter. LAB and yeast isolates that were co-inoculated with toxigenic *A. flavus* La3228 resulted in an over 50% reduction in the aflatoxin B1 concentration. The aflatoxin reduction in isolates retrieved from maize performed better than that in those retrieved from millet. The isolate *L. plantarum* M26 of millet origin resulted in a 62% and 63% reduction in aflatoxin B1 and B2 concentrations. *L. fermentum* W310, from white maize, resulted in an 86% and 75% reduction in aflatoxins B1 and B2. Similarly, the aflatoxin concentrations were lowered by the yeast *C. tropicalis* MY115 of millet origin, which resulted in a 60% and 77% reduction in aflatoxins B1 and B2. Whereas *C. tropicalis* YY25, which was isolated from yellow maize, resulted in a 60% and 31% reduction in aflatoxins B1 and B2. Compared to the yeast isolates, the *L. plantarum* and *L. fermentum* performed better in terms of reducing the aflatoxin B1 concentrations than the aflatoxin B2 concentrations. The yeast *C. tropicalis* MY115 had a greater ability at reducing aflatoxin B2 compared to B1; however, this was not the case for *C. tropicalis* YY25 (Table 4).

#### 2.6.2. Amino Acid Profiles without the Inclusion of Probiotics

In the absence of probiotics, there were differences in the amino acid concentrations among systems with no *A. flavus* (control), toxigenic *A. flavus* and atoxigenic *A. flavus*. Compared to the control, the majority of the amino acid concentrations (13/17) were significantly higher in the presence of the toxigenic *A. flavus* when compared to the control. This differed only with histidine, tyrosine, lysine, and isoleucine, where their concentrations were similar. On the contrary, the majority of the amino acid (13/17) concentrations were similar between *ogi* containing the atoxigenic *A. flavus* 3279 and the control, differing with glycine, proline, cystine, and leucine. The cystine concentrations were higher (*p* < 0.05) when the atoxigenic *A. flavus* 3279 was included, but the concentrations were lower for the other three amino acids.

#### 2.6.3. Amino Acid Concentrations with the Inclusion of LAB and *Aspergillus*

There were differences in the amino acid profiles depending on whether the fermenting organism was *Lm. fermentum* or *Lp. plantarum*. In the systems containing toxigenic *A. flavus*, there were higher (*p* < 0.05) concentrations of threonine, tyrosine, and methionine, and lower concentration (*p* < 0.05) of lysine in the *Lm. fermentum* systems compared to the *Lp. plantarum* systems (*p* < 0.05). In the systems containing atoxigenic *A. flavus*, there were higher (*p* < 0.05) concentrations of serine and alanine, and lower (*p* < 0.05) concentrations of leucine in the *Lm. fermentum* systems compared to the *Lp. plantarum* systems.

#### 2.6.4. Amino Acid Concentrations with the Inclusion of Yeasts and *Aspergillus*

The amino acid profiles differed depending on whether the *C. tropicalis* (MY115) strain (isolated from millet) or YY25 (isolated from maize) were used. In the systems containing toxigenic *A. flavus*, there were higher (*p* < 0.05) concentrations of cysteine, tyrosine, valine, methionine, and leucine, and lower (*p* < 0.05) concentrations of lysine in the *C. tropicalis* MY115 systems compared to the YY25 systems. In the systems containing atoxigenic *A. flavus*, there were lower (*p* < 0.05) concentrations of aspartic acid, serine, glutamine, arginine, alanine, proline, tyrosine, valine, leucine, and phenylalanine, and there were higher (*p* < 0.05) cysteine concentrations when *C. tropicalis* MY115 was used compared to when YY25 was used.

#### 2.6.5. Amino Acid Concentrations with the Inclusion of Probiotics, Excluding *Aspergillus*

In systems lacking *Aspergillus* but containing probiotics, there were differences in 7/17, 3/17, 3/17, and 1/17 amino acids compared to the control (lacking probiotics or Aspergilli) in the *Lm. fermentum*, *C. tropicalis* MY115, *C. tropicalis* YY25, and *Lp. plantarum* systems, respectively. In the *Lm. fermentum* systems, amino acids with differing concentrations (glycine, threonine, arginine, alanine, proline, methionine, and phenylalanine) were elevated (*p* < 0.05), whereas in the *Lp. plantarum* systems, the glycine levels were reduced (*p* < 0.05). In the *C. tropicalis* MY115 systems, glycine had lower concentrations, whereas the methionine and phenylalanine concentrations were elevated. In the *C. tropicalis* YY25 systems, glycine and alanine had lower concentrations and leucine had elevated concentrations.

## 3. Discussion

Aflatoxins in cereals sometimes go undetected because of the loose testing systems in open markets, which are common in many developing countries. This has implications whereby aflatoxins enter the diets of several members of the population, which has been reported in the literature [5,22,23]. It is critical for policies and institutional frameworks to be in place to prevent the entry of aflatoxins into the diets of populations in regions where aflatoxin contamination is endemic. However, it is not always possible to completely exclude the entry of aflatoxins, and sometimes, the concentrations may occur in low levels. There are no tolerable limits of aflatoxins, and aflatoxins are to be as low as reasonably possible. So, whereas aflatoxins are regulated at concentrations as low as 4 ppb in the EU, achieving non-detectable limits is preferable. While efforts such as reducing aflatoxin levels through several methods including the use of bioprotectants continue to be encouraged [5,24], the use of fermentation as a biological method can further remove aflatoxins from the food matrices where they occur. Other methods for post-exposure alleviation include the use of animal function regulators, such as curcumin [25]. Fermentation offers this benefit in addition to providing other organoleptic benefits to the consumer and easing digestion for weaning children and convalescents [17].

From the current study, the identification of LAB and yeasts as probiotics with beneficial properties of reducing aflatoxin concentrations was identified. Ascertaining the safety of these organisms was critical as it is important to ensure that the decontamination of dietary exposure to aflatoxins is not compounded by the use of unsafe organisms to ensure continued food safety. The identified organisms are generally regarded as safe “(GRAS)”. The rate of multiplication of these probiotic microorganisms was about 10 times more in maize than in sorghum and millet, suggesting that the former was a preferable nutrient source for the microbes than the latter. This is probably due to the harder external testa of sorghum and millet compared to maize, which has a softer testa. Several LAB and yeasts were identified. Among the LAB, *Lm. fermentum* and *Lp. plantarum* were identified as more beneficial for the reduction in aflatoxins and had beneficial probiotic potentials in their abilities to survive the gastrointestinal environments based on in vitro observations and their safety. Among the yeasts, two isolates of *C. tropicalis* were the most beneficial based on similar attributes [5,17].

The acidification test indicates that the microorganisms were good acidifiers, while the bile salt tolerance was conducted on the LAB and yeast strains to determine their ability to possess good characteristics that are necessary for conferring health benefit to humans. Probiotics are viable, non-pathogenic microorganisms (bacteria and yeasts), which when ingested in sufficient numbers, can confer health benefits to the host. The viability of these microorganisms upon ingestion and their survival through the gastrointestinal tract is critical to confer health benefits to the host [26].

The capability of the LAB and yeasts to produce acid with a stronger reduction in the pH of the medium of growth qualifies them as good candidates for the starter culture fermentation procedure because they will outcompete other microorganisms and their ability to tolerate different concentrations of bile salts qualifies their probiotic candidacy [17,27,28,29]. The selected strains possessed the potentials to reduce the pH in vitro to pH 5.0 or less after 24 h and 48 h, respectively, indicating that they are good acidifiers. These qualities had been reported in some LAB and yeasts strains obtained from traditional fermented food products [23,28,29,30]. Bile salt tolerance is a precondition for the colonization and metabolic activities of the microorganisms in the small intestine of the host [23,28,30]. Their tolerance to bile salts was assessed because bile is a lipid emulsifying agent that is released into the duodenum after food intake and has a potential antimicrobial activity [24]. In the selection of probiotics, a bile salt concentration of 0.3% is critical in their determination as potential probiotics, but the bile salt concentration of the human intestine could increase to as high as 0.5% [27,29,31]. The LAB and yeast strains in this study showed tolerance up to a 1% bile salt concentration, with variations in their survival patterns. The survival ability of these strains in the presence of bile signified that they could endure the acidic gastric environment within the small intestine of the host’s digestive system, thereby imparting their benefits.

All the fermented samples contained no detectable sugars at the completion of the fermentation period (results not shown). Although there were differences in sugar utilization patterns among the probiotics, the sugar utilization patterns of *Lm. fermentum* and *Lp. plantarum* were similar (Table 1). There were, however, differences in the amino acid utilization content in the matrices that could have occurred due to specific probiotic or specific *A. flavus* utilization (or biosynthesis) patterns, as this varied depending on whether an aflatoxigenic or non-aflatoxigenic fungus was present.

Atoxigenic *Aspergillus* are utilized in aflatoxin biological control, and as such, the grains harvested from such treated fields would be associated with these fungi. This study demonstrated that this biocontrol treatment does not significantly influence the nutritional composition in fermentation systems as majority of the amino acid (13/17) concentrations were comparable to the control (*p* > 0.05) when the atoxigenic *A. flavus* 3279 was present. However, in the presence of aflatoxigenic *A. flavus*, the amino acid concentrations were largely dissimilar (13/17 amino acids) (*p* < 0.05). Others have suggested that amino acids are important for aflatoxin dose classification systems that are lacking probiotics [32]. The current study suggests that this could remain the case in probiotic fermentation systems.

The serine concentrations were higher (*p* < 0.05) than the concentrations in the control and atoxigenic systems, suggesting that these were more involved in the secondary metabolism associated with the toxigenic *Aspergillus* (Table 5) than the other amino acids. However, the proline and leucine concentrations were lower (*p* < 0.05) than the control when atoxigenic *A. flavus* was present, suggesting the importance of these amino acids in the primary metabolism. 

In both the toxigenic and atoxigenic fermentation systems with LAB, *Lm. fermentum* resulted in higher amino acid concentrations compared to the *Lp. plantarum* systems. This suggests that *Lm. fermentum* would be more beneficial for nutritive purposes. However, *C. tropicalis* strains showed differences with MY115, resulting in higher amino acid concentrations in the presence of the toxigenic mold and YY25 fermentations resulting in higher concentrations in the presence of the atoxigenic mold.

There were observed differences in the extent of their abilities to detoxify *ogi* containing aflatoxins based on contamination by a highly aflatoxigenic *A. flavus* isolate. All four of the selected probiotics were able to reduce the aflatoxin concentrations by more than 50%. However, some were better at the detoxification of aflatoxin B1 compared to aflatoxin B2. Among the major aflatoxins B1, B2, G1, and G2, aflatoxin B1 is the most toxigenic and carcinogenic. *Lm. fermentum* reduced the aflatoxin B1 concentrations by 86%, whereas *Lp. plantarum* reduced the concentration of aflatoxin B1 by 61.6%, indicating that this *Lm. fermentum* isolate may be more beneficial for food safety purposes regarding the extent of decontamination and the greater ability at detoxifying B1 compared to the *Lp. plantarum* isolate used. This is indicative of inter-species differentiation in their abilities to detoxify aflatoxins [23]. For the yeast probiotics isolated, although both isolates were *C. tropicalis* isolates, the isolate of maize origin *C. tropicalis* YY25 performed better in terms of the percentage reduction in aflatoxin B1 and the greater ability to detoxify aflatoxin B1 compared to the *C. tropicalis* MY115 isolate of millet origin. This suggests that within species, there can be strain differentiation in their abilities to detoxify aflatoxins and the extent to which they are able to detoxify specific aflatoxins. However, it is not possible to deduce from the current study whether the origin of the isolate is a determinant of its detoxification ability.

Studies indicate that the decontamination of aflatoxins occurs by binding, with suggestions that biochemical processes may be involved [33]. Although this study did not aim to investigate the mode of decontamination, there are indications that the secondary metabolism continued with the toxigenic strain, as seen from larger deviations in amino acid baselines compared to the atoxigenic strain. It is plausible that decontamination occurs via binding. Additionally, the differences in aflatoxin detoxification are both intraspecies- and interspecies-specific. Consequently, to improve nutritive value and safety, a combination-fermentation system would provide value in both the detoxification and improvement in the digestibility and nutritive value. It would also be interesting to understand how the organoleptic properties may be affected by single- and combination-fermentation systems. This should be explored in future research.

## 4. Conclusions

The results from this study demonstrates the utilization of probiotics including lactic acid bacteria and yeasts in aflatoxin decontamination. High deviations in the amino acid profiles were observed in the presence of the lactic acid bacteria and yeasts, suggesting that the secondary metabolism of aflatoxins was not largely impeded during the fermentation process. The decontamination process must be investigated further to improve our understanding.

## 5. Materials and Methods

### 5.1. Samples Collection

Three kilograms each of maize, millet, and sorghum *ogi* were purchased from Wakajaye, Ibadan (GPS coordinates: 7.42118, 3.97637), from market vendors. Samples were collected in clean plastic containers and transported at ambient temperature to the Microbiology Laboratory of the University of Ibadan for isolation of yeasts and lactic acid bacteria. Thereafter, isolate selection was conducted based on their probiotic potentials, as determined by their acidification, their resistance to bile salts, and their safety assessments. These are described in more detail as follows.

### 5.2. Isolation and Characterization of Lactic Acid Bacteria

From each of the different samples (*ogi* from white maize, yellow maize, millet, and sorghum), 25 isolates of lactic acid bacteria were collected. Briefly, 5 mL of *ogi* and 45 mL of sterile distilled water were combined and homogenized to make a stock solution. From the stock solution, serial dilutions were prepared and inoculated on deMann Rogosa Sharpe (MRS) agar for differential isolations of LAB. Isolation was performed using pour plate method and was incubated for 48 h [34]. Isolates were thereafter sub-cultured by repeated streaking to obtain pure isolates, and pure isolates were kept in MRS broth and glycerol at 4 °C until required.

Gram staining, catalase, oxidase, potassium hydroxide (KOH), and sugar fermentation tests were performed. Pure cultures were macroscopically examined for their pigmentation, colony shape, and elevation. These were conducted for their morphological, physiological, and biochemical classifications, as described by Fawole and Oso [35]. Isolated LAB were later assessed for their probiotic potential, as described hereafter.

### 5.3. Isolation and Characterization of Yeasts

From each of the different samples (*ogi* from white maize, yellow maize, millet, and sorghum), 25 isolates of yeasts were collected. From the stock solution prepared, serial diluted samples were inoculated using pour plate method in malt extract agar (MEA) supplemented with chloramphenicol (100 mg L^−1^) to preclude bacterial growth. Inoculated plates were incubated for 72 h. Thereafter, isolates were streaked repeatedly to obtain pure isolates. Pure isolates were observed for color, appearance, elevation, and margin, and were then stored at 4 °C in MEA slants until required. Morphological, physiological, and biochemical tests were also conducted, including microscopic observations of colonies stained with lactophenol blue and observed at ×40 objective lens for cell shape and occurrence and position of buds, as described by Alakeji et al. [36]. Then, the yeast isolates were assessed for their probiotic potentials.

### 5.4. Determination of Probiotic Potentials of the LAB and Yeast Isolates

The probiotic potentials of the isolates were determined based on in vitro assessments, such as the acidification potential and resistance to bile salts. These are hereafter described.

*Acidification*: Yeasts and LAB were screened for their acidification potential as a signal to their fermentation ability, as described by Banwo et al. (2013). Briefly, isolates (1%) were inoculated in 10 mL of sterile MRS and ME broth with pH adjusted to 6.54 and incubated for 6, 24, and 48 h in covered test tubes at 30 °C. Isolates with superior acidification potentials, as assessed by the pH reduction in the broth, were pre-selected.

*Resistance to bile salt*: Tolerance to bile salts is critical due to the presence of bile salts in the digestive system. Therefore, the bile salt tolerance was determined as previously described [27,29]. Briefly, LAB and yeast strains (2% *v/v*) were inoculated in MRS broth and ME broth, respectively, containing 0.3, 0.5, and 1% (*w/v*) of bile salt. These were then incubated at 37 °C for 24 h. Optical density measurements were obtained using a spectrophotometer Jenway 6350 UV/Visible Spectrophotometer, (Jenway, Staffordshire, United Kingdom) at 560 nm. An isolate’s tolerance to bile salts was determined based on their optical density measurements in comparison to the control culture that did not contain bile salts. The growth was expressed in percentage.

### 5.5. Safety Assessment of the Pre-Selected LAB and Yeasts

Pre-selected strains were assessed for their safety for use as a starter culture in a controlled fermentation process. To determine these, the strains were subjected to hemolysis, DNAse, and gelatinase assessments.

Firstly, hemolysis by LAB and yeast cells was determined by incubating 24 h old LAB and yeast cells collected from MRS and ME broths, respectively, on blood agar plates supplemented with 5% defibrinated sheep blood (Thermos Fisher Scientific, United Kingdom) for 24 to 48 h. Cells without a clear zone (γ-hemolysis) were considered safe, while cells that produced α-hemolysis (partial hydrolysis) or β-hemolysis (a clear zone around the bacteria and yeast growth) were considered unsafe [37].

Secondly, DNase assessments involved spot inoculation of pre-selected isolates on DNase (CM0321B, Oxoid, Basingstoke, United Kingdom) agar plates, followed by a 48-h incubation. Plates were then flooded with concentrated HCl, and the presence of a clear zone indicated production of DNase by LAB and yeast cells and so they were considered unsafe, while safe cells did not produce a clear zone (which indicated that the cells could not breakdown DNA) [38].

Finally, ability to hydrolyze gelatin was determined by inoculating 24 h old cells in test tubes containing nutrient gelatin medium (6 g of gelatin and 0.65 g of nutrient broth in 50 mL of distilled water) and incubating them at 35 °C for 48 h and then chilling at 4 °C for 15 to 20 min to enable gelling. When the control sample gelled, the samples were left to stand for 20 min and then tilted to identify hydrolysis of gelatin in comparison to the control. Cells that hydrolyzed gelatin were considered unsafe and were not selected [39,40].

### 5.6. Molecular Characterization of the Selected LAB and Yeast Isolates

Following putative identifications of isolates based on their morphological, physiological, and biochemical characteristics, isolate characterization, and then the pre-selections based on their superior abilities as probiotics and safety, selected isolate identities that were further confirmed genotypically. Two LAB (M26 from millet and W310 from white maize) and two yeast isolates (MY115 from millet and YY25 from yellow maize) were sequenced. Briefly, cell genomic DNA was extracted using Zymo research quick DNA mini preparation kit as per manufacturer’s instructions, and target gene segments were amplified by polymerase chain reactions (PCR). The LAB and yeast strains were identified by amplification and sequencing of the partial 16S rDNA gene and ribosomal internal transcribed spacer (ITS) region, employing primers designated as 27 F (5′-GAGTTTGATCCTGGCTCAG-3′) and 1492R (5′-TACCTTGTTACGACTT-3′) for LAB and ITS 4 (5′-CCTCCGCTTATTGATATGC-3′) and ITS 5 (5′–GGA AGTAAA AGT CGTAAC AAG G–3′) for the yeast, respectively [41,42,43]. The strains were sequenced at the Department of Biosciences and Biotechnology, International Institute of Tropical Agriculture (IITA), Ibadan. The species designation was obtained by employing the Basic Logarithmic Alignment Search Tool (BLAST) algorithm in National Centre for Biotechnology Information (NCBI) database. Sequences with 100% similarity were regarded as belonging to the same taxonomy group. The sequences obtained were deposited to the GenBank database (https://www.ncbi.nlm.nih.gov/genbank/, accessed on 30 January 2023) and accession numbers were assigned for public accessibility.

### 5.7. Controlled Fermentation with LAB and Yeasts in the Presence of Toxigenic and Atoxigenic A. flavus Strains

*Collection and preparation of A. flavus strains*: Two *A. flavus* strains, La 3228 (toxin-producing) and La 3279 (non-toxin-producing), were sourced from the Pathology and Mycotoxin Laboratory of the International Institute of Tropical Agriculture collection. Isolates were grown on potato dextrose agar (PDA) in 90 mm Petri dishes by dropping silica granules to which the isolates had been adsorbed and incubating at 25 °C for 7 days. Thereafter, the isolates were sub-cultured and incubated similarly to enhance spore production. Post-incubation, spores were rinsed from the Petri dishes with sterile distilled water containing 0.1% (*v/v*) Tween 80 solution, and the spore suspension was adjusted to 1 × 10 ^6^ spores mL^−1^.

*Collection and preparation of probiotic strains*: Two LAB strains, W310 (from white maize) and M26 (from millet), and two yeast strains, MY115 strain (from millet) and YY25 strain (from yellow maize), were retrieved from storage, as previously mentioned. The LAB strains were incubated in MRS broth for 24 h at 37 °C. Thereafter, the broth was diluted to 10^−8^ and inoculated on MRS agar in triplicates and incubated at 37 °C for 48 h. At the completion of the incubation period, cells were rinsed in sterile distilled water containing 0.1% Tween 80 solution. Yeast cells were prepared similarly to the LAB suspensions, except for the use of ME broth rather than MRS broth.

After probiotic LAB and yeasts were selected, they were assessed for their abilities to detoxify aflatoxins in a controlled fermentation process with maize grains. The controlled fermentation process was conducted using white maize grains (TZL COMP.4 DT C3 WHITE) with undetectable baseline aflatoxin levels, which was verified using the method described by Agbetiameh et al. [44].

*Fermentation process*: White maize was milled and sterilized by autoclaving at 121 °C for 30 min at 15 psi. Thereafter, five grams of milled maize was combined with 10 mL of sterile distilled water to make *ogi* slurry and was inoculated in different treatment combinations, indicated in Table 6. Yeasts and LAB suspensions (10^8^ cfu mL^−1^) were used with 1 mL of *Aspergillus* in their respective treatment combinations. After inoculation, samples were mixed and left to ferment at ambient temperature for five days, and then, aflatoxin and amino acid concentrations were quantified. The CAMAG TLC Scanner 3 using winCATS 1.4.2 (CAMAG, AG, Muttenz, Switzerland) was used for aflatoxin measurements, and amino acid concentrations were quantified by HPLC. Aflatoxin quantification was as subsequently described.

Prior to analysis of amino acids, samples were lyophilized and then pulverized to obtain a stable and uniform matrix for amino acid profiling. Briefly, 0.05 g of each sample was hydrolyzed with boiling 6 M HCl in a 10 mL hydrolysis tube within CEM Microwave Discover Workstation. Thereafter, the samples were centrifuged at 1500 rpm for 15 min to obtain a clear hydrolysate.

HPLC quantification of amino acids was performed using Waters Alliance 2695 HPLC system with a 2475 Multi λ Fluorescence detector (Waters, Milford, MA, USA). Excitation was at 250 nm and emission at 395 nm and separation occurred in the AccQ•Tag amino acid column Nova-Pak C 18, 4 µm (150 × 3.9 mm). Column was conditioned at 37 °C, with injection volume of 10 μL and concentration of amino acids from 2.5 to 250 pmol. A gradient mobile phase was used for chromatography. The mobile phase consisted of Eluent A (prepared from Waters AccQ•Tag Eluent A concentrate, by adding 200 mL of concentrate to 2 L of Milli-Q water and mixing), Eluent B (acetonitrile, HPLC grade), and Eluent C (Milli-Q water) [45].

### 5.8. Aflatoxin Quantification

Aflatoxins B1, B2, G1, and G2 were analyzed as previously described by others [44,46]. Briefly, 20 g each per replicate was blended with 100 mL of 70% methanol using a Waring blender. Thereafter, the suspension was agitated for 30 min at 400 rpm and the suspension was filtered through Whatman No. 1 filter paper. The filtrate was collected into 250 mL separating funnels, to which 20 mL of distilled water and 25 mL of dichloromethane was added. This was mixed gently and left for 10–15 min to allow for separation of the matrix. The dichloromethane phase was filtered through a bed of 20 g anhydrous sodium sulfate, which was contained in fluted Whatman No. 4 filter paper. Subsequently, the filtrate left to evaporate in a fume hood and residues were dissolved in 1 mL dichloromethane. Homogenates were directly spotted 4 µL alongside aflatoxin standards on thin layer chromatography (TLC) aluminum (20 × 10 cm) silica gel 60 F plates and developed with diethyl ether–methanol–water (96:3:1) chamber. Plates were visualized under ultraviolet light (365 nm) for presence or absence of aflatoxins and later quantified with a scanning densitometer (CAMAG TLC Scanner 3) and quantification software (win CATS 1.4.2, Camag, AG, Muttenz, Switzerland).

### 5.9. Statistical Analysis

Percentage reduction in aflatoxin concentration (AF conc.) for aflatoxin B1 and aflatoxin B2 was determined using the following formula:
[(AF conc. of *ogi* and *A. flavus* 3228 − AF conc. of *ogi* and *A. flavus* 3228 and probiotic)]/(AF conc. of *ogi* and *A. flavus* 3228) × 100%(1)

The mean concentrations of amino acids were analyzed for statistical differences using analysis of variance (ANOVA) and separated using Student–Newman–Keuls test (at α = 0.05). Analysis was conducted using SAS v9.1 (SAS Institute, Cary, NC, USA).

## Figures and Tables

**Figure 1 toxins-15-00210-f001:**
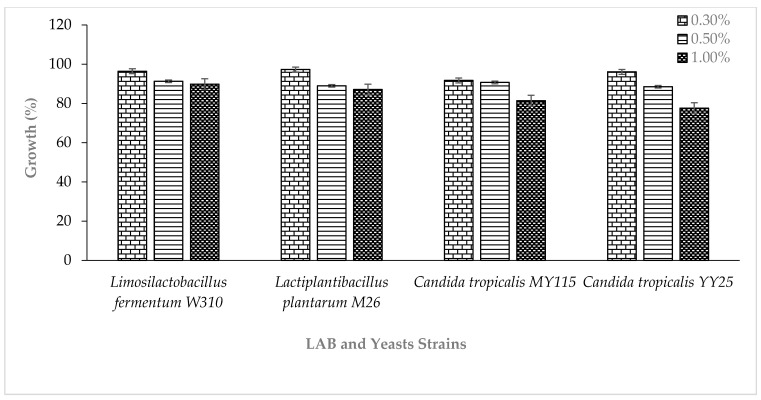
Tolerance of the selected LAB and yeast strains to 0.30, 0.50, and 1.00% bile salt concentration.

**Table 1 toxins-15-00210-t001:** Sugar and sugar alcohol utilization patterns of lactic acid bacteria.

LAB	Compounds Able to Utilize	Compounds Unable to Utilize	Occurrence (%)	Types of *Ogi* Isolated from
*Lb. acidophilus*	Glucose, galactose, fructose, d-sorbitol, and lactose	Sucrose and mannitol	5%	Maize (yellow and white)
*Lb. brevis*	Glucose, galactose, fructose, d-sorbitol, and lactose	Sucrose and mannitol	8%	Millet, sorghum, and yellow maize
*Lc. casei*	Glucose, sucrose, galactose, fructose, mannitol, lactose, and d-sorbitol (Y16 only)	d-sorbitol, sucrose (Y16 only) and mannitol (Y16 and Y42 only), d-sorbitol (Y42 only), and galactose (Y42 only)	10%	Maize (yellow and white), millet, and sorghum
*Lb. delbrueckii*	Glucose, fructose, d-sorbitol, and lactose	Sucrose, galactose, and mannitol	17%	Maize (yellow and white), millet, and sorghum
*Lm. Fermentum*	Glucose, sucrose, galactose, fructose, lactose, and mannitol (Y310)	d-sorbitol and mannitol	15%	Maize (yellow and white), millet, and sorghum
*Lb. leichmanii*	Glucose, sucrose, fructose, and lactose	Galactose, d-sorbitol, and mannitol	3%	Millet and sorghum
*Lb. paraplantarum*	Glucose, galactose, d-sorbitol, Lactose (W44 only), and fructose (W44 only)	Sucrose, fructose, mannitol, and lactose	5%	Maize (yellow and white)
*Lb. pentosus*	Glucose, galactose, fructose, d-sorbitol, and lactose	Sucrose and mannitol	8%	Millet and sorghum
*Lp. plantarum*	Glucose, sucrose, galactose, fructose, and mannitol	d-sorbitol, lactose, and mannitol (W37)	12%	Maize (yellow and white), millet, and sorghum
*Leuc. mesenteroides*	Glucose, sucrose, galactose, and lactose	Fructose, D-sorbitol, and mannitol	17%	Maize (yellow and white), millet, and sorghum

**Table 2 toxins-15-00210-t002:** Sugar and sugar alcohol utilization patterns of yeasts.

Yeasts	Compounds Able to Utilize	Compounds Unable to Utilize	Occurrence (%)
*Candida krusei*	Glucose	Sucrose, galactose, fructose, d-sorbitol, mannitol, and lactose	35%
*C. tropicalis*	Glucose and sucrose, galactose and fructose,	Mannitol and lactose	35%
*Geotrichum candidum*	Glucose and fructose	Sucrose, galactose, d-sorbitol, mannitol, and lactose	30%

**Table 3 toxins-15-00210-t003:** Acidification and safety assessments of the LAB and yeast strains.

LAB/Yeast Strains	0 h	6 h	24 h	48 h	Hemolysis	Gelatinase	DNAse
*Limosilactobacillus fermentum* W310	6.54	5.92	3.78	3.70	γ	-	-
*Lactiplantibacillus plantarum* M26	6.54	5.48	4.26	3.65	γ	-	-
*Candida tropicalis* MY115	6.54	5.85	4.27	4.26	γ	-	-
*Candida tropicalis* YY25	6.54	5.90	5.00	4.60	γ	-	-

- = negative, γ = gamma hemolytic reaction (no hemolysis), and h = hour.

**Table 4 toxins-15-00210-t004:** Percentage (%) reduction in aflatoxin in *ogi* after 5 days of controlled fermentation.

	Aflatoxin Conc in ng/g				Perc. Reduction	
Treatments	B_1_	B_2_	G_1_	G_2_	B_1_	B_2_
*Ogi* alone	0	0	0	0	NA	NA
*Ogi* and *L. plantarum* M26	0	0	0	0	NA	NA
*Ogi* and *L. fermentum* W310	0	0	0	0	NA	NA
*Ogi* and *C. tropicalis* MY115	0	0	0	0	NA	NA
*Ogi* and *C. tropicalis* YY25	0	0	0	0	NA	NA
*Ogi* and *A. flavus* 3279 (atoxigenic)	0	0	0	0	NA	NA
*Ogi* and *A. flavus* 3228 (toxigenic)	7048	309	0	0	NA	NA
*Ogi*, *Lp. plantarum* M26, and *A. flavus* 3279	0	0	0	0	NA	NA
*Ogi*, *Lp. plantarum* M26, and *A. flavus* 3228	2704	113	0	0	62	63
*Ogi*, *Lm. fermentum* W310, and *A. flavus* 3279	0	0	0	0	NA	NA
*Ogi*, *Lm. fermentum* W310, and *A. flavus* 3228	965	77	0	0	86	75
*Ogi*, *C. tropicalis* MY115, and *A. flavus* 3279	0	0	0	0	NA	NA
*Ogi*, *C. tropicalis* MY115 and *A. flavus* 3228	2827	70	0	0	60	77
*Ogi*, *C. tropicalis* YY25, and *A. flavus* 3279	0	0	0	0	NA	NA
*Ogi*, *C. tropicalis* YY25, and *A. flavus* 3228	2835	212	0	0	60	31

Keys: AFB1 = aflatoxin B1, AFB2 = aflatoxin B2, AFG1 = aflatoxin G1, AFG2 = aflatoxin G2, *A. flavus* = *Aspergillus flavus*, *Lp. plantarum* = *Lactiplantibacillus plantarum*, *Lm. fermentum* = *Limosilactobacillus fermentum*, and *C. tropicalis* = *Candida tropicalis*. NA = not applicable.

**Table 5 toxins-15-00210-t005:** Amino acid concentration in maize under different fermentation treatments.

Treatments	Asp	Ser	Gln	Gly	His	Thr	Arg	Ala	Pro	Cys	Tyr	Val	Met	Lys	Ile	Leu	Phe
*Ogi* alone	0.65 C	0.18 DE	0.74 CD	0.32 D	0.32 BCD	0.1 DE	0.05 D	0.3 CDE	0.39 C	0.06 D	0.11 DEF	0.15 BC	0.01 DE	0.49 C	0.14 B	0.14 D	0.08 CD
*Ogi* + 3228 (toxigenic)	1.99 B	1.61 B	8.15 B	1.12 B	0.77 AB	1.4 A	1.77 AB	2.34 B	2.68 A	7.27 A	0.15 DE	1.65 A	0.08 C	0.49 C	1.06 B	3.63 A	1.28 B
*Ogi* + 3279 (atoxigenic)	0.69 C	0.12 E	0.72 CD	0.12 EF	0.13 CD	0.36 CD	0.02 D	0.39 CD	0.09 D	0.15 C	0.09 DEF	0.11 BC	0.01 DE	0.46 C	0.17 B	0.02 E	0.01 D
*Ogi* + *Candida tropicalis* (millet *ogi)*	0.7 C	0.18 DE	0.75 CD	0.15 E	0.19 BCD	0.22 DE	0.08 D	0.27 CDE	0.22 CD	0.08 D	0.12 DEF	0.08 BC	0.09 C	0.49 C	0.15 B	0.1 D	0.15 B
*Ogi* + *Candida tropicalis* (millet *ogi*) + 3228	2.78 A	2.00 A	9.45 A	1.51 A	0.82 AB	1.64 A	2.05 A	2.98 A	3.09 A	0.68 B	1.73 A	2.05 A	0.35 B	0.85 B	4.4 A	0.1 D	1.4 B
*Ogi* + *Candida tropicalis* (millet *ogi*) + 3279	0.64 C	0.05 FG	0.54 D	0.08 A	1.23 A	0.61 BC	1.12 C	0.24 DEF	0.39 C	0.17 C	0.04 F	0.04 C	0.02 DE	0.59 C	0.24 B	0.13 D	0.14 CD
*Ogi* + *Candida tropicalis* (yellow maize *ogi*)	0.55 C	0.17 DE	0.74 CD	0.02 G	0.02 D	0.03 E	0.03 D	0.12 F	0.2 CD	0.06 D	0.09 DEF	0.08 BC	0.01 DE	0.51 C	0.13 B	0.14 D	0.54 C
*Ogi* + *Candida tropicalis* (yellow maize *ogi*) + 3228	2.78 A	1.94 A	8.89 AB	1.34 AB	0.71 AB	1.59 A	1.97 A	2.66 AB	2.71 A	0.07 D	1.32 B	0.14 BC	0.14 C	1.22 A	4.48 A	0.02 E	1.56 B
*Ogi* + *Candida tropicalis* (yellow maize *ogi*) + 3279	1.92 B	1.43 C	7.96 B	1.13 AB	0.72 AB	0.93 B	1.59 B	2.43 B	2.61 A	0.06 D	1.22 B	1.43 A	0.03 D	0.46 C	1.79 B	1.07 B	1.2 B
*Ogi* + *Limosilactobacillus fermentum*	0.54 C	0.15 DE	0.97 C	0.7 C	0.57 ABC	2.03 A	1.15 C	2.68 AB	0.93 B	0.06 D	0.06 EF	0.41 B	0.49 A	0.46 C	0.1 B	0.26 C	3.83 A
*Ogi* + *Limosilactobacillus fermentum* (yellow maize *ogi*) + 3228	0.68 C	0.08 F	0.65 CD	0.01 G	0.03 D	0.68 BC	0.02 D	0.36 CDE	0.08 D	0.06 D	0.3 C	0.07 BC	0.13 C	0.07 D	0.07 B	0.09 D	0.02 CD
*Ogi* + *Limosilactobacillus fermentum* (yellow maize *ogi*) + 3279	0.7 C	0.18 DE	0.75 CD	0.03 G	0.04 CD	0.05 E	0.05 D	0.45 C	0.29 CD	0.06 D	0.11 DEF	0.16 BC	0.0 E	0.59 C	0.14 B	0.14 D	0.12 CD
*Ogi* + *Lactiplantibacillus plantarum*	0.71 C	0.23 D	0.76 CD	0.07 EFG	0.13 CD	0.03 E	0.01 D	0.2 EF	0.27 CD	0.06 D	0.11 DEF	0.14 BC	0.01 DE	0.45 C	0.14 B	0.12 D	0.11 CD
*Ogi* + *Lactiplantibacillus plantarum* (millet *ogi*) + 3228	0.68 C	0.02 F	0.68 CD	0.01 G	0.03 D	0.11 DE	0.03 D	0.43 C	0.25 CD	0.06 D	0.19 D	0.13 BC	0.01 DE	0.58 C	0.1 B	0.13 D	0.1 CD
*Ogi* + *Lactiplantibacillus plantarum* (millet *ogi*) + 3279	0.6 C	0.06 F	0.47 CD	0.05 FG	0.14 CD	0.15 ED	0.09 D	0.15 F	0.36 CD	0.06 D	0.09 DEF	0.1 BC	0.01 DE	0.57 C	0.09 B	1.09 B	0.09 CD

Means with different letters along the column are statistically different from one another at *p* < 0.05. Square root-transformed means were analyzed by ANOVA and separated using Student–Newman–Keuls test using SAS version 9.4. Asp = aspartic acid, Ser = serine, Gln = glutamine, Gly = glycine, His = histidine, Thr = threonine, Arg = arginine, Ala = alanine, Pro = proline, Cys = cystine, Tyr = tyrosine, Val = valine, Met = methionine, Lys = lysine, Ile = isoleucine, Leu = leucine, and Phe = phenyleanaline.

**Table 6 toxins-15-00210-t006:** Treatments for investigating influence of lactic acid bacteria and yeast during controlled fermentation of *ogi* in the presence of *Aspergillus*.

Treatments
*Ogi* alone
*Ogi* + *Lactiplantibacillus plantarum* M26 *Ogi* + *Limosilactobacillus fermentum* W310
*Ogi* + *Candida tropicalis* MY115 *Ogi* + *Candida tropicalis* YY25
*Ogi* + La3279 *Ogi* + La3228
*Ogi* + *Lactiplantibacillus plantarum* M26 + La3279 *Ogi* + *Limosilactobacillus fermentum* W310 + La3279
*Ogi* + *Candida tropicalis* MY115 + La3279 *Ogi* + *Candida tropicalis* YY25 + La3279
*Ogi* + *Lactiplantibacillus plantarum* M26 + La3228 *Ogi + Limosilactobacillus fermentum* W310 + La3228
*Ogi* + *Candida tropicalis* MY115 + La3228 *Ogi* + *Candida tropicalis* YY25 + La3228

## Data Availability

According to Institutional requirements, data will be archived within IITA’s open access platform.

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
