# Peer review of "Detoxification of Aflatoxins in Fermented Cereal Gruel (*Ogi*) by Probiotic Lactic Acid Bacteria and Yeasts with Differences in Amino Acid Profiles"

_toxins, 2023, doi:10.3390/toxins15030210_

Round 1
Reviewer 1 Report (Previous Reviewer 1)
This manuscript is truly interesting and presents a very well-designed strategy to demonstrate the degradation efficiency of aflatoxins in fermented cereal gruel (ogi) by probiotic lactic acid bacteria and yeasts with differences in amino acid profiles. The modification of the manuscript basically meets the acceptance standard. But some minor details need to be addressed:
1. This needs minor corrections. Please ensure the format of B1, B2, G1, G2 in Table 5 is uniform.
2. The reference have a few formatting problems, please adjust it.
Author Response
This manuscript is truly interesting and presents a very well-designed strategy to demonstrate the degradation efficiency of aflatoxins in fermented cereal gruel (ogi) by probiotic lactic acid bacteria and yeasts with differences in amino acid profiles. The modification of the manuscript basically meets the acceptance standard. But some minor details need to be addressed:
- This needs minor corrections. Please ensure the format of B1,B2, G1, G2 in Table 5 is uniform.
Thank you. This has been effected.
- The reference have a few formatting problems, please adjust it.
Thank you. This has been formatted to journal's requirements.
Reviewer 2 Report (Previous Reviewer 3)
The author's have been made significant changes to the manuscript but still need few more corrections, please proofread the document and make all headings look same (including sub headings).
The section on statistical analysis in the "Methods" section is absent, which would have been beneficial for the readers to understand the reasoning and approach taken for handling the data.
Author Response
Reviewer 1
The author's have been made significant changes to the manuscript but still need few more corrections, please proofread the document and make all headings look same (including sub headings).
Answer: Thank you. This has been done.
The section on statistical analysis in the "Methods" section is absent, which would have been beneficial for the readers to understand the reasoning and approach taken for handling the data.
Answer: We thank the reviewer for this comment. We have included a section for statistical analysis in the methods section.
Reviewer 3 Report (Previous Reviewer 2)
The manuscript determined characterization and probiotic potentials of the LAB and yeast isolates and investigated changes to amino acid concentrations during fermentation with probiotics in the presence of aflatoxigenic strain or atoxigenic strain. The work is meaningful and substantial. However, some comments need to addressed before the manuscript can be accepted:
1. Lines 159-160: primers 27F and 1492R, please present them in sequence of 5’ end to 3’ end instead of that of 3’to 5’, which will be more in line with writing format conventions.
2. Three section of “2.7. Fermentation of maize with LAB and yeast in the presence of toxigenic and atoxigenic A. flavus”, “2.8. Controlled fermentation process of maize grains with probiotics in the presence of A. flavus” “2.9. Controlled fermentation process” should be combined into one section.
3. Line 198: “108” lack of unit. Please add unit, 108 CFU ML-1?
4. Line 262: please add the punctuation mark “,” after the word “white”.
5. Lines 267-268: all isolates were classified into five genera, which genera do the ten isolates in the Table 2 belong to? Give information about the genera of these isolates in the Table 2.
6. Lines 286-287: Utilization of fructose was 68%. But occurrence frequency of yeasts, C. tropicalis and Geotrichum candidum, was 65% (35%+30%), check them please.
7. Lines 309-312: the sentence elucidates the bile salt tolerance of four LAB and yeasts strains (97.3%, 89.8%, 96.1% and 81.4% respectively) to two concentrations (0.3% and 1.0%), which is confused me. Which concentration are these percentages corresponding to, 0.3% or 1.0%?
8. Line 356: “iso-leucine” should be “isoleucine.
9. Lines 402-407: the prevention of fungal growth and/or aflatxoin control using several methods was introduced. Except for the detoxification means of aflatxoins introduced herein, other detoxifying ways, such as animal function regulator (curcumin), were not introduced. It is necessary to briefly introduce the detoxifying means in this paragraph, and allow me to suggest the following publication to be cited herein, please.
Wang, Y.; Liu, F.; Zhou, X.; Liu, M.; Zang, H.; Liu, X.; Shan, A.; Feng, X. Alleviation of Oral Exposure to Aflatoxin B1-Induced Renal Dysfunction, Oxidative Stress, and Cell Apoptosis in Mice Kidney by Curcumin. Antioxidants 2022, 11, 1082. https://doi.org/10.3390/antiox11061082.
10. Line 412: add the punctuation mark “.” after “(GRAS)”.
11. Line 424: “which indicates that they good acidifiers”, rewrite the sentence due to grammar problems please.
12. In Table 6, please check whether Glu is the abbreviation of glutamic acid or glutamine (as shown in line 472, Glu=Glutamine). According to international general rules, the abbreviation of glutamate should be Glu, while glutamine should be Gln. In addition, standard abbreviation forms of methionine and isoleucine should be Met and Ile.
13. Line 471: “Tyr=Throline” should be “Thr=Threonine”
14. The Conclusion needs to be improved. For instance, it is more appropriate move the sentence “It is plausible that decontamination occurs via binding” to the section of Discussion. Binding ability of the isolated probiotics on the aflatoxin was not determined in this study, so binding mechanism should not be concluded, but should only be discussed.
Author Response
Reviewer 2
The manuscript determined characterization and probiotic potentials of the LAB and yeast isolates and investigated changes to amino acid concentrations during fermentation with probiotics in the presence of aflatoxigenic strain or atoxigenic strain. The work is meaningful and substantial. However, some comments need to addressed before the manuscript can be accepted:
- Lines 159-160: primers 27F and 1492R, please present them in sequence of 5’ end to 3’ end instead of that of 3’to 5’, which will be more in line with writing format conventions.
Answer: Thank you. This has been done.
- Three section of “2.7. Fermentation of maize with LAB and yeast in the presence of toxigenic and atoxigenic A. flavus”,“2.8. Controlled fermentation process of maize grains with probiotics in the presence of A. flavus” “2.9. Controlled fermentation process” should be combined into one section.
Answer: Thank you. This combination has been effected.
- Line 198: “108” lack of unit. Please add unit, 108CFU ML-1?
Answer: Thank you. This correction has been effected.
- Line 262: please add the punctuation mark “,” after the word “white”.
Answer: Thank you. This correction has been effected.
- Lines 267-268: all isolates were classified into five genera, which genera do the ten isolates in the Table 2 belong to? Give information about the genera of these isolates in the Table 2.
Answer: This has been indicated in Table 2. Thank you.
- Lines 286-287: Utilization of fructose was 68%. But occurrence frequency of yeasts, C. tropicalisand Geotrichum candidum, was 65% (35%+30%), check them please.
Answer: This has been corrected. Thank you.
- Lines 309-312: the sentence elucidates the bile salt tolerance of four LAB and yeasts strains (97.3%, 89.8%, 96.1% and 81.4% respectively) to two concentrations (0.3% and 1.0%), which is confused me. Which concentration are these percentages corresponding to, 0.3% or 1.0%?
Answer: This has been corrected to indicate that these tolerance percentages were for 0.3% bile salt concentration. Thank you.
- Line 356: “iso-leucine” should be “isoleucine.
Answer: This has been corrected. Thank you.
- Lines 402-407: the prevention of fungal growth and/or aflatxoin control using several methods was introduced. Except for the detoxification means of aflatxoins introduced herein, other detoxifying ways, such as animal function regulator (curcumin), were not introduced. It is necessary to briefly introduce the detoxifying means in this paragraph, and allow me to suggest the following publication to be cited herein, please.
Wang, Y.; Liu, F.; Zhou, X.; Liu, M.; Zang, H.; Liu, X.; Shan, A.; Feng, X. Alleviation of Oral Exposure to Aflatoxin B1-Induced Renal Dysfunction, Oxidative Stress, and Cell Apoptosis in Mice Kidney by Curcumin. Antioxidants 2022, 11, 1082. https://doi.org/10.3390/antiox11061082.
Answer: This has been included. Thank you.
- Line 412: add the punctuation mark “.” after “(GRAS)”.
Answer: This has been included. Thank you.
- Line 424: “which indicates that they good acidifiers”, rewrite the sentence due to grammar problems please.
Answer: This has been corrected. Thank you.
- In Table 6, please check whether Glu is the abbreviation of glutamic acid or glutamine (as shown in line 472, Glu=Glutamine). According to international general rules, the abbreviation of glutamate should be Glu, while glutamine should be Gln. In addition, standard abbreviation forms of methionine and isoleucine should be Met and Ile.
Answer: This has been corrected. Thank you.
- Line 471: “Tyr=Throline” should be “Thr=Threonine”
Answer: This has been corrected. Thank you.
- The Conclusion needs to be improved. For instance, it is more appropriate move the sentence “It is plausible that decontamination occurs via binding” to the section of Discussion. Binding ability of the isolated probiotics on the aflatoxin was not determined in this study, so binding mechanism should not be concluded, but should only be discussed.
Answer: This has been corrected. Thank you.
Reviewer 4 Report (New Reviewer)
The article examines a significant problem: the contamination of traditional African food - ogi - with aflatoxins. The authors took a very interesting approach: to show the detoxication process by comparison of toxigenic and non-toxigenic Aspergillus strains in their behavior.
Significant results have been obtained and the work can be published after correction.
1. The introduction should be more detailed. For example, to describe ogi: what it is and how it is prepared (https://doi.org/10.3390/fermentation9020111).
In addition, a recent review revealed the potential of lactobacilli to perform mycotoxin detoxification which should also be mentioned (https://doi.org/10.3390/nu14102038).
2. As a major remark, I would like to mention the presentation of the newly isolated lactobacilli and yeast strains.
Page 8, line 297: a comparison should be made to the type strains of the specified species. The presented % of similarity should be with the sequence of the type strain. Also, it's good to show the sequences of all isolates, not only percentages of species. Twenty-five lactobacillus strains and 25 yeast strains may be best displayed on phylogenetic trees, especially since they were identified by a classical genetic method.
3. Figure 1 is too large for its content. Please correct the ordinate by adding tick marks.
4. Table 2: "Freq" - to be written in full. Might as well replace it with "% of isolates" if that's what's meant.
5. Unify "ml" "mL", and "L" throughout the text.
6. After the first mention of a species, it may be referred to only by its first letter in the text, or by an appropriate abbreviation, for example, Latilactobacillus - as La. "Aspergilli" should be replaced by Aspergillus.
7. Citations in the text and references should be fundamentally rearranged. The references must be in the order of the first mention in the text, and in the text - they must be numbered. Each reference must be followed by a link. Please see the journal requirements.
Author Response
Reviewer 3
The article examines a significant problem: the contamination of traditional African food - ogi - with aflatoxins. The authors took a very interesting approach: to show the detoxication process by comparison of toxigenic and non-toxigenic Aspergillus strains in their behavior.
Significant results have been obtained and the work can be published after correction.
- The introduction should be more detailed. For example, to describe ogi: what it is and how it is prepared (https://doi.org/10.3390/fermentation9020111).
In addition, a recent review revealed the potential of lactobacilli to perform mycotoxin detoxification which should also be mentioned (https://doi.org/10.3390/nu14102038).
Answer: Thank you. These references have been included in the introduction and discussion sections to provide more information to the readers on ogi preparation and mycotoxin detoxification.
- As a major remark, I would like to mention the presentation of the newly isolated lactobacilli and yeast strains.
Page 8, line 297: a comparison should be made to the type strains of the specified species. The presented % of similarity should be with the sequence of the type strain. Also, it's good to show the sequences of all isolates, not only percentages of species. Twenty-five lactobacillus strains and 25 yeast strains may be best displayed on phylogenetic trees, especially since they were identified by a classical genetic method.
Answer: Thank you very much for this comment. We did not carry out molecular characterization on all the 50 isolates obtained in the study. We screened the isolates and chose the best performing for the molecular characterisation as displayed in Table 3 with their accession numbers assigned to them for easy access on the GenBank database (https://www.ncbi.nlm.nih.gov/genbank/). Further details can be checked on the above stated website. The nucleotide sequences are added as a supplementary file.
- Figure 1 is too large for its content. Please correct the ordinate by adding tick marks.
Answer: Thank you. The tick marks have been added to the Figure and the figure reduced in size.
- Table 2: "Freq" - to be written in full. Might as well replace it with "% of isolates" if that's what's meant.
Answer: Thank you. This has been changed to occurrence (%).
- Unify "ml" "mL", and "L" throughout the text.
Answer: Thank you. This has been unified.
- After the first mention of a species, it may be referred to only by its first letter in the text, or by an appropriate abbreviation, for example, Latilactobacillus - as La. "Aspergilli" should be replaced by Aspergillus.
Answer: Thank you. This has been done.
- Citations in the text and references should be fundamentally rearranged. The references must be in the order of the first mention in the text, and in the text - they must be numbered. Each reference must be followed by a link. Please see the journal requirements.
Answer: Thank you. This has been done.
Round 2
Reviewer 3 Report (Previous Reviewer 2)
All my comments have been addressed in the revised manuscript, and I think it is suitable for the publication in the journal of "Toxins".
Reviewer 4 Report (New Reviewer)
I am fully satisfied with the changes and improvements.
The manuscript can be published in its current form.
This manuscript is a resubmission of an earlier submission. The following is a list of the peer review reports and author responses from that submission.
Round 1
Reviewer 1 Report
The manuscript“Probiotic lactic acid bacteria and yeasts detoxify aflatoxins during fermentation of cereal gruel (ogi), with differences in aminoacid profiles when fermented with toxigenic or atoxigenic Aspergillus flavus ” is lack of experimental integrity. Although a solid piece of work is presented in the manuscript, the work is incremental and does not present significant advances in current scientific knowledge.
1. The title is too long and lacks conciseness
2. The manuscript does not make it clear, what is the mechanism by which probiotic strains are used to degrade aflatoxin? Whether it inhibits the growth of toxin-producing strains or metabolizes toxin molecules.
3. Introduction clarifies the focus of this study to assess the safety of natural probiotic yeast and lactic acid bacteria in ogi, but is relatively brief in the subsequent discussion.
4. The deviation in total amino acid concentration used in the abstract to show the effect of antidotes on the metabolism of toxic strains is too general and does not take a single amino acid perspective.
5. The manuscript does not consider whether aflatoxin has an effect on the growth of probiotic lactic acid bacteria and yeast.
6. The discussion section is too cumbersome, and it is recommended that it be simplified and condensed to facilitate understanding.
7. In Table 4 ,a revised model appears on page 7.
8. In Table 5 ,the content is not displayed completely on page 11.
Reviewer 2 Report
Susceptibility of multiple cereals to mycotoxin contamination including aflatoxins seriously pose threat to human and animal health. This study screened probiotic lactic acid bacteria and yeasts with detoxify aflatoxin function during fermentation of cereal gruel (ogi). The study work is meaningful and designs well. However, some comments need to addressed before the manuscript can be accepted as follows”:
Aflatoxin has multiple toxic effects on humans and animals, but was not introduced in the “Introduction” section. It should be briefly introduced the toxic effects, and related studies such as the following reports should be cited herein.
Jin, S.; Yang, H.; Jiao, Y.; Pang, Q.; Wang, Y.; Wang, M.; Shan, A.; Feng, X. Dietary Curcumin Alleviated Acute Ileum Damage of Ducks (Anas platyrhynchos) Induced by AFB1 through Regulating Nrf2-ARE and NF-κB Signaling Pathways. Foods 2021, 10, 1370. https://doi.org/10.3390/foods10061370
Pickova, D.; Ostry, V.; Toman, J.; Malir, F. Aflatoxins: History, Significant Milestones, Recent Data on Their Toxicity and Ways to Mitigation. Toxins 2021, 13, 399. https://doi.org/10.3390/toxins13060399
Lines 79-82:numbers of power value of 10 should be Superscript form, correct them please.
Line 102: add “.” after “millet” please.
In table 2: “2L. delbrueckii”, delete “2” please.
The title of 2.5 “Changes to aflatoxin concentrations and amino acid profile due to LAB and yeast fermentation” should be corrected, for example, changed to “aflatoxin concentration changes due to LAB and yeast fermentation”. In addition, “2.5.1 Aflatoxin concentration changes” should be deleted.
Line 155: “there” should be changed to “that”.
Lines 158-160: as elucidated herein, over 50% reduction in aflatoxin concentration was reduced. But C. tropicalis YY25 only resulted in 31% reduction in aflatoxins B2 as shown in table 4. Correct it please.
Lines 160-167: Significant digits of aflatoxin reduction percentage used in these sentences were not consistent with those presented in table 4. Unify them please.
Lines 179-192: the first sentence and the last sentence in this paragraph should be deleted because the two sentences presented the content of probiotic fermentation that has no relation with the title of 2.6.1 “Amino acid profiles without the inclusion of probiotics”. In addition, table 5 should be mentioned in this paragraph.
Lines 201-203: “the concentrations of glycine, histidine, arginine, alanine, proline, leucine and phenylalanine were lower (p<0.05) than amino acid concentrations in the absence of Aspergilli”, among these amino acids, His was not different between “ogi+Lactobacillus fermentum” (0.57ABC) and “ogi+Lactobacillus fermentum+3279” (0.04CD) as shown in table 5. Correct it, please.
Lines 214-218: the second sentence in this paragraph presented that four amino acids were different ogi containing C. tropicalis isolated from millet and that isolated from maize, but the third sentence presented all amino acids except for Phe in ogi containing C. tropicalis from millet was higher (p<0.05) that those in ogi containing C. tropicalis from maize. Please explain this contradiction.
Many sentences about the description of statistical differences were confusing, especially in those of lines 219-231. In these sentences, it is difficult to identify which group is used as the control group for comparison. For example, as elucidate in line 220, there were lower concentration (p < 0.05) of glycine, but there was no significant difference between “Ogi+Candida tropicalis (millet ogi) + 3228” (1.51A) and “Ogi + Candida tropicalis (millet ogi) + 3279” (0.08A), both of which with the same uppercase letter“A”. I am also confused that there is no difference between the two values of 1.5 and 0.08.
In table 5, Presentation form of the amino acid name should be unified. Some were in full name, such as Aspartic acids (should be Aspartic acid), Alanine, Proline, but some were in abbreviations, such as Ser, Glu, Gly. In addition, part of table 5 is outside of page.
Section of “conclusion” needs to be added in this manuscript.
Line 347: add “.” after “(Akin-Osanaiye and Kamalu, 2019)”, please.
Line 349: “at 4 oC” should be “at 4 ℃”.
Line 353: “physiological.” delete “.” please.
Line 363: “at 4 oC” should be “at 4 ℃”.
Lines 375 and 381: “at 30 oC” should be “at 30 ℃”.
Line 375, LAB was incubated in covered test tubes at 30 ℃, I consider that the temperature is lower for LAB incubation. Are the authors sure the temperature was adopted”
Line 402: delete “ ” please.
Line 420: “35 oC” and “4 oC “ should be “at 35 ℃” and “4 ℃”.
Line 454: “106”should be “106”
Line 460: “10 -8” should be “10 -8”
Line 474: “108” should be “108”
Line 492: “injection volume was 10 μl) was the injection volume”, rewrite the sentence please.
Line 496: what’s meaning of “6. Patents”, it should be deleted.
Reviewer 3 Report
The research is interesting, and the manuscript needs changes.
1. Introduction – Too lengthy and can be cut down and straighter to the point.
2. Mentioned the importance but not clearly mentioned about how they will do it.
3. Grammatical mistakes
4. Line 65-66 Redundant cut down to 2 sentences
5. Line 41 Missing reference
6. Line 42 Missing reference
Results
7. 2.1 These results are not clear. Sort these values in a table format.
8. Line 79 – Formatting errors
9. Line 81- 96-h try to be consistent
10. 2.2 Formatting errors
Discussion
11. Line 233- 244 Reference, check grammar
12. There is no mentioning of the role of amino acids in the introduction, the results and discussion is mostly about amino acids.
Methods:
13. 5.2 and 5.3 what are the quality controls that were taken during the isolation and characterization of yeast and LAB?
14. 5.8 Formatting errors
15. Line 479 Aflatoxin quantification was as previously described- where? No citation
16. Line 488 HPLC quantification – Does method adopted from literature or standard method?
17. Table 5 Cannot see the complete table. Change the layout.